# Knowledge, Awareness, and Attitude of Healthcare Stakeholders on Alzheimer’s Disease and Dementia in Qatar

**DOI:** 10.3390/ijerph20054535

**Published:** 2023-03-03

**Authors:** Pradipta Paul, Ziyad Riyad Mahfoud, Rayaz A. Malik, Ridhima Kaul, Phyllis Muffuh Navti, Deema Al-Sheikhly, Ali Chaari

**Affiliations:** 1Weill Cornell Medicine-Qatar, Doha 24144, Qatar; 2Division of Medical Education, Weill Cornell Medicine-Qatar, Doha 24144, Qatar; 3Division of Epidemiology, Department of Population Health Sciences, Weill Cornell Medicine, NY 10065, New York, USA; 4Division of Medicine, Weill Cornell Medicine-Qatar, Doha 24144, Qatar; 5Division of Cardiovascular Sciences, Faculty of Biology, Medicine and Health, University of Manchester, Manchester M13 9NT, UK; 6Faculty of Science and Engineering, Manchester Metropolitan University, Manchester M15 6BH, UK; 7Division of Continuing Professional Development, Weill Cornell Medicine-Qatar, Doha 24144, Qatar; 8Premedical Division, Weill Cornell Medicine-Qatar, Doha 24144, Qatar

**Keywords:** proficiency, competence, ability, geriatrics, continuing medical education, neurodegenerative disease, practice, knowledge

## Abstract

Dementia is characterized by progressive cognitive decline, memory impairment, and disability. Alzheimer’s disease (AD) accounts for 60–70% of cases, followed by vascular and mixed dementia. Qatar and the Middle East are at increased risk owing to aging populations and high prevalence of vascular risk factors. Appropriate levels of knowledge, attitudes, and awareness amongst health care professionals (HCPs) are the need of the hour, but literature indicates that these proficiencies may be inadequate, outdated, or markedly heterogenous. In addition to a review of published quantitative surveys investigating similar questions in the Middle East, a pilot cross-sectional online needs-assessment survey was undertaken to gauge these parameters of dementia and AD among healthcare stakeholders in Qatar between 19 April and 16 May 2022. Overall, 229 responses were recorded between physicians (21%), nurses (21%), and medical students (25%), with two-thirds from Qatar. Over half the respondents reported that >10% of their patients were elderly (>60 years). Over 25% reported having contact with >50 patients with dementia or neurodegenerative disease annually. Over 70% had not undertake related education/training in the last 2 years. The knowledge of HCPs regarding dementia and AD was moderate (mean score of 5.3 ± 1.5 out of 7) and their awareness of recent advances in basic disease pathophysiology was lacking. Differences existed across professions and location of respondents. Our findings lay the groundwork for a call-to-action for healthcare institutions to improve dementia care within Qatar and the Middle East region.

## 1. Introduction

Dementia is characterized by acquired, progressive impairment in multiple cognitive domains, including memory, that interfere with independence in social or occupational function [1,2]. It leads to disability and dependability in the elderly and is a paramount source of distress among affected individuals, their families, and caregivers, [3] making it a major public health priority [4]. Worldwide, at least 57.4 million people live with dementia and 1.6 million succumb to it annually [5,6]. While the proportion of the global population aged ≥ 65 years will approximately double between 2019 and 2050, the prevalence of dementia and Alzheimer’s disease (AD), accounting for 60–70% of dementias, are projected to reach 152.8 million and 106.4 million, respectively [5,7,8]. However, there is marked geographical heterogeneity in these projections of dementia prevalence. While an increase of 53% is expected in the high-income Asia Pacific countries, this rise is expected to be 367% in the Middle East and North Africa (MENA) region, with marked increases in Saudi Arabia (898%), Oman (943%), Bahrain (1084%), and the United Arab Emirates (1795%). Qatar, with a population close to 3 million, seems to be especially at risk; from only 4201 cases of dementia in 2019, the prevalence of dementia is estimated to increase by 1926% to 85, 046 by 2050, largely fueled by aging of the population—the single biggest risk factor for AD [5,9]. Additionally, Qatar also has a high incidence of cardiometabolic risk factors, such as hyperglycemia, smoking, hypertension, and obesity to name a few, all of which are known to increase risk of dementia [10,11,12,13]. Table 1 provides further context regarding dementia care in Qatar [4,7,14,15,16,17,18,19,20,21,22,23].

The increasing prevalence of dementia requires a simultaneous increase in the number of health care professionals (HCPs) providing competent care for this population. Adequate knowledge and competence of HCPs affects timing of diagnosis, implementation of intervention, and quality of care, which impact overall patient outcomes [24,25,26,27,28]. Increased knowledge among HCPs reduces stigma, increases patient quality of life, and reduces caregiver burden, and the converse is also true [29,30,31]. Studies have linked confidence and quality of dementia management ability to not only previous formal dementia education, but also current knowledge, awareness, and attitude of HCPs [32,33,34]. Formally, measuring these factors by surveying this population helps identify gaps in current healthcare delivery and promotes effective resource allocation for professional development via continuing education programs that refresh old knowledge and introduce new advances in the field [35]. To our knowledge, details of the knowledge, attitude, and awareness of HCPs on dementia and AD in the region are not present in literature. This survey has addressed:The gaps in current knowledge on AD and dementia, among major health care stakeholders (physicians, nurses, dentists, pharmacists, allied health professionals, medical students, educators, and researchers) in the region.Their attitude towards addressing knowledge gaps via continuing medical education (CME) webinars.

## 2. Materials and Methods

The current pilot cross-sectional survey targeted various major healthcare stakeholders, including physicians, nurses, dentists, pharmacists, allied health professionals (AHPs), medical students, educators, and researchers, among others. It aimed to assess the knowledge, awareness, and attitude of such individuals towards AD and dementia to determine the needs of a CME webinar series addressing needs-based regional challenges.

### 2.1. Measures

A short, online, self-contained survey in English was designed by the authors in collaboration with local and international researchers in the field of neurodegenerative diseases, locally based physicians who routinely diagnose and manage such patients, and institution-affiliated medical education experts who regularly design and deliver professional development programs and lectures for healthcare professionals in the region. We also adapted components from the published and previously tested Alzheimer’s Disease Knowledge Scale (ADKS) [36], the Alzheimer’s Disease Awareness Scale (ADAS) [37], and the Dementia Attitudes Scale (DAS) [38], in addition to incorporating questions to specifically assess the needs of healthcare practitioners in the region.

The structured questionnaire (Appendix A) contained questions to record sociodemographic data including participant’s age, occupation, and their primary country of practice/study/work, questions to assess their experience in caring for elderly and people with AD, and whether they had any recent participation in AD or dementia-related educational training, including webinars, training courses, or grand rounds. Occupation was recorded as follows: physician, nurse, dentist, pharmacist, social worker, AHPs, student, administrator, researcher, educator, insurance representative, and others. To assess current experience and exposure to treating patients with dementia, we inquired about the proportion of the respondents’ current patients who were over 60 years of age, and how many patients with dementia were being seen in a healthcare setting annually.

Additionally, the main questionnaire consisted of seven questions to assess knowledge, nine questions to assess awareness/attitude, and five miscellaneous questions to assess awareness of the pathophysiology and scope of new treatments for dementia. In the 21 questions assessing the main outcome of knowledge, attitude, or awareness, participants selected answers based on a five-point Likert scale ranging from ”strongly disagree” to ”strongly agree” or from “not at all aware” to “extremely aware”. An overall index of knowledge was calculated by summing up the correct answers.

### 2.2. Process

The survey was circulated for four weeks between 19 April and 16 May 2022 via e-mail to over 9742 healthcare practitioners, academics, researchers, and other professionals that were subscribed to the mailing list of the Weill Cornell Medicine-Qatar continuing professional development (CPD) division, which routinely delivers continuing education opportunities to HCPs in the region. It should be noted that professionals based both within and outside Qatar were recipients of this mail and thus survey respondents (Appendix A). The survey was hosted on Qualtrics XM software and mass e-mails containing a link to the survey were circulated; follow-up organic advertisement was done via social media platforms and private messaging. Random, lottery-based financial incentives via gift vouchers were promised to five random respondents to increase response rate. Ethical approval for the study was granted by the Weill Cornell Medicine-Qatar Institutional Review Board (IRB#: 22-00013) as a low-risk study. All individual level collected data was confidential and shared only amongst the authors after removal of personal identifiers. We also reviewed the literature for similar quantitative surveys globally and in the MENA region.

### 2.3. Data Analysis

Survey responses were downloaded onto Microsoft Excel sheets and analyzed using IBM SPSS software. Age- and practice-related variables were summarized using frequency distribution. For each of the seven knowledge questions, the percentage of participants who correctly answered “strongly agree” or “moderately agree” was computed for each of the five professional groups. Also, a total of correct answers was computed as a score out of seven with one for correct answer and zero for an incorrect answer. The score was summarized using mean ± standard deviation, along with the median, the interquartile range, and the minimum and maximum values attained by the participants. For each of the “attitudes” questions, participant attitudes were considered positive by combing those who answered “disagree” or “strongly disagree” for some questions and “agree” and “strongly agree” for other questions depending on the wording of the questions. The percentage of participants with positive attitude was computed for each question and stratified by the five profession groups. As for awareness questions, participants were considered aware based on answering “moderately” or “extremely” aware for that question. The proportion of awareness for each question was computed and stratified by the profession groups. No statistical analysis was done to compare the five groups as this is a pilot study, and with such numbers, it is not powered for such comparisons.

### 2.4. Review Methodology

In order to provide context and compare the results of the current study, we finely reviewed existing literature for similar quantitative surveys across the MENA region, with further representation from similar studies outside this region for comparison. We searched PubMed and Google Scholar for terms such as “Middle East”, “dementia”, “Alzheimer’s”, “knowledge”, “attitude”, and “awareness” up to December 2022. We particularly reviewed and summarized studies investigating the knowledge/attitude/awareness/practice of the general population and healthcare professionals towards AD and dementia, whereas we integrated similar studies outside this region into the discussion.

## 3. Results

### 3.1. Demographics and Characteristics of Participants

Of 9742 email recipients, 3206 opened the email, 794 clicked on the relevant survey link, and ultimately a total of 229 complete responses (overall response rate of 2.35%) to the questionnaires was recorded. Table 2 summarizes the participants’ sociodemographic data and relevant experience in the field. Most respondents primarily identified as either physicians (20.5%), nurses (21.0%), students (24.5%), and educators/researchers (15.3%), while the rest (18.8%) reported being dentists, pharmacists, or allied healthcare professionals, among others. Almost half (47.6%) the respondents were between 31–50 years of age, largely stemming from the large nurse and educator/researcher population (79.5% and 60.5%, respectively) of the same age group. Most physicians (61.7%) tended to be older (41–60 years), whereas all students were between 17–30 years of age. Overall, 154 (67.2%) respondents reported having their primary place of practice/study in Qatar, with 72.9% of nurses and 89.3% of students reporting the same. Regarding their patient characteristics, 102 (55.6%) participants reported that more than 10% of their current patients were elderly (>60 years of age), among whom, 47 (20.5%) reported this proportion to be greater than 50%. Over a quarter (26.7%) of participants reported seeing patients with neurodegenerative diseases (≥50 patients annually) in their practice or study. Importantly, however, less than one in three (29.7%) respondents reported having any training in dementia/neurodegenerative disease in the last 2 years.

### 3.2. Knowledge Regarding Alzheimer’s Disease and Dementia

Overall, the respondents demonstrated moderate knowledge regarding Alzheimer’s disease and its manifestations, with the mean (SD) score being 5.3 ± 1.5 out of 7 (median [IQR]: 5.0 [5.0–6.0]; range 0.0–7.0) (Table 3). This average was 5.7 ± 1.3 among physicians, 5.4 ± 1.4 among students, 5.2 ± 1.4 among educators/researchers, 5.0 ± 1.5 among nurses, and 5.0 ± 1.8 among other professions. Most (72.1%) correctly identified that loss of memory and inability to perform daily tasks by the elderly require a medical consultation, whereas only 42.8% identified that changes in executive functioning and balancing finances were not physiologically expected in the elderly. Most (79.0%) did not believe AD to be a result of psychological distress or physical injury, and 88.6% agreed that early AD detection could result in a better response to treatment. Only 68.1% of respondents believed that artificial intelligence may be of benefit in neurodegenerative diseases, whilst most believed that lifestyle shapes the brain (86.5%) and agreed that physical activity, sleep, and cognitive function were related in the elderly (90.8%).

### 3.3. Attitude Regarding Alzheimer’s Disease and Dementia

The awareness/attitude of respondents towards Alzheimer’s disease and its manifestations are displayed in Table 4. Overall, 68.6% of participants disagreed with hiding a diagnosis from relatives with AD and 80.8% did not believe it was best for AD patients to avoid social interactions to avoid embarrassment to themselves; these figures did not vary significantly between responses of those from different professions. On the other hand, only 57.6% of respondents disagreed with turning to alternative medicine when a relative developed signs/symptoms of dementia, with physicians (72.3%), educators/researchers (62.8%), and students (60.7%) being more disapproving compared to nurses (45.8%) and other professions (46.5%). Most believed that when a patient with AD develops difficulty performing everyday tasks, the judiciary should save the patient’s rights, with only 23.6% disagreeing. A total of 77.3% of respondents denied feeling embarrassed if a close relative was diagnosed with AD and an even greater proportion (85.6%) reported that they would not deny a diagnosis of AD in a relative. A total of 63.3% of respondents disagreed with having patients with AD being looked after in state nursing homes rather than at home. Regarding research and expertise of healthcare professions, 87.8% believed in the potential of dementia research to improve the outlook for patients, families, and providers, and 86.9% considered it important that HCPs should be aware of the most recent updates in the field of neurodegenerative disease and dementia.

### 3.4. Awareness of Pathophysiology and Understanding of New Advancements in Neurodegenerative Diseases

Responses to questions on pathophysiology and experimental technologies in dementia are displayed in Table 5. Less than half (46.7%) reported being either moderately or extremely aware of the pathophysiological mechanisms behind the common neurodegenerative diseases, with the lowest awareness among nurses (37.5%), other professions (41.9%), and students (42.9%), whereas physicians (61.7%) and educators/researchers (51.4%) displayed greater confidence. Less than half (43.2%) of participants reported being either moderately or extremely aware of the significance of protein misfolding and amyloid formation, a key hallmark of various neurodegenerative diseases. The greatest awareness was among educators/researchers (62.9%), whilst the least awareness was among nurses (12.5%). Less than one third (31.9%) were either moderately or extremely aware of the applications of artificial intelligence in daily life, whilst a similar proportion (32.3%) reported being either moderately or extremely aware of its potential application in healthcare. Finally, less than one in five (18.8%) had moderate or extreme awareness of the potential of corneal confocal microscopy in the diagnosis of peripheral neuropathies and central neurodegenerative diseases, although his proportion was greater among physicians (27.7%) and educators/researchers (31.4%).

### 3.5. Differences in Responses in Those in Qatar vs. Outside Qatar

Overall, 154 (67.2%) respondents were based in Qatar, with 53.2% of physicians, 72.9% of nurses, 89.3% of students, 42.9% of researchers/educators, and 67.4% of respondents from other professions reporting the same (Table 2). Interestingly, there were a few notable differences (Figure 1). Only 67.5% of those based in Qatar (versus 81.3% of those based outside Qatar) disagreed that symptoms of loss of memory were normal in the elderly and do not require medical attention. Moreover, only 37.7% of Qatar responses (versus 53.3% external) believed that deterioration in daily planning and financial independence is not expected naturally in the elderly. A lesser proportion of respondents in Qatar disagreed that AD may result from black magic or psychological distress (74.7%), and indicated that they would resort to alternative forms of medicine (51.9%), compared to those based outside Qatar (88.0% and 69.3%, respectively). Interestingly, 20.1% of respondents in Qatar did not believe it is necessary for the judiciary to protect AD patients’ rights if they have difficulty performing daily tasks, whereas 30.7% of those based outside Qatar thought so. A higher proportion of Qatar-based respondents (80.5% vs. 70.7%) believed that they would not feel embarrassed if a close relative was diagnosed with AD. Notably, fewer Qatar-based respondents disagreed with having AD patients being admitted to state nursing homes rather than remaining at home (59.1% vs. 72.0%). Other notable differences were in respect to disease pathophysiology and perception of the potential of AI and CCM in medicine. Generally, a lesser proportion of those based in Qatar were either moderately or extremely aware of the mechanisms behind neurodegenerative diseases (42.9% vs. 54.7%), the significance of protein misfolding and amyloid formation (36.4% vs. 57.3%), the current and future role of AI in daily life (29.2% vs. 37.3%) and medicine (29.3% vs. 37.3%), and of CCM in diagnosis of peripheral neuropathies and central neurodegenerative diseases (14.9% vs. 26.7%).

### 3.6. Literature Review of Similar Studies in the MENA Region

Dementia represents a significant challenge not only in Qatar, but also in the rest of the MENA region. Sociocultural and political factors may limit research and dissemination of knowledge. Promoting a knowledge-based culture by imbibing traditional perspectives with evidence-based models across the populous, in contrast to the currently prevalent stigma and lack of awareness, should be a strategy to optimize dementia care [39]. In Table 6 we present the results of a literature review summarizing important findings and conclusions of contemporary studies in the Middle East investigating the knowledge, attitude, or awareness of the general public (n = 11) and healthcare practitioners (n = 8) on dementia and Alzheimer’s disease through quantitative survey methods, with one prior study based in Qatar.

## 4. Discussion

As the prevalence of dementia and dementia-causing neurodegenerative disease increases with the continuously aging population, attention is shifting towards early diagnosis to optimize long-term patient outcomes. Challenges in detecting and managing mild cognitive impairment (MCI), often the first presentation of dementia, revolve around a triad of hesitant patients, unprepared providers, and misaligned environments [58]. In order to improve dementia care, addressing simultaneous challenges faced by all involved healthcare stakeholders is necessary and incomplete without the others. This study is the first of its kind in Qatar that has determinedthe the knowledge, awareness, and attitudes of current and future healthcare professionals towards dementia and its associated neurodegenerative diseases. In addition, determining the willingness of HCPs to engage, improve knowledge, and reduce stigma will in turn help us identify effective and efficient areas of intervention, for example, through CME/CPD programs, independent webinars, group discussions, and interactive sessions. This study will help inform future, large-scale studies directed not only at healthcare professionals, but also the general public, in order to address the increasing challenges in dementia care in the region.

### 4.1. Knowledge, Awareness, and Attitude of the General Population

As AD and dementia increasingly contribute to the global burden of disease, researchers have investigated whether public knowledge and beliefs have consistently been followed and kept up in order to promote help-seeking behavior and reduce associated stigma. In Australia, Smith et al. [59] showed that many did not hold beliefs or have knowledge that would otherwise reduce dementia risk; only 41.5% of respondents of a large public quantitative survey believed that dementia risk could be reduced. Recent data from Macau in China revealed that although older adults had more dementia knowledge, they had less favorable attitudes when compared to the youth [60]. Wu et al. [60] apply the concept of construal level theory to conclude that bridging the existing psychological distance of dementia via intergenerational programs can increase awareness among younger adults. A large cross-sectional investigation of public attitude towards dementia in Bristol and South Gloucestershire in the UK revealed that individuals who were younger, identified themselves as White, and with personal experience of dementia (among close family/friends) had a more positive attitude than their counterparts [61]. A systematic review of 38 studies investigating public awareness about preventative dementia treatment revealed that nearly half considered dementia to be a normal and non-preventable part of aging, with the role of cardiovascular risk factors being poorly understood, although awareness improved over time [62].

Regions with high migrant populations may have more challenges posed by linguistic and cultural differences, translating to obstacles in identification, assessment, and diagnosis in the clinic, which may especially apply to Qatar. Sagbakken et al. [63] reveal that two major misconceptions from these patients and their families include considering some symptoms to be attributable to normal aging or something to be ashamed of. Monsees et al. [64] show that migrant patients’ willingness to use services increased after incorporating their culture into an aspect of care, which increased comfort, utilization, and satisfaction in this group. In a study from Copenhagen, involving native Danish, Polish, Turkish, and Pakistani immigrants, the latter two groups were more likely to hold normalizing and stigmatizing views of AD which were not significantly influenced by education or acculturation levels [65]. Thus, ethnic background may be strongly associated with wrong or misguided knowledge and perception of dementia and AD, leading to challenges in accessing healthcare services in such populations. Even within a region, knowledge and perceptions regarding dementia/AD may differ between different ethnic groups, which is especially relevant for Qatar with its exceptional diversity [65].

### 4.2. Knowledge, Awareness, and Attitude of Healthcare Professionals and the Effect of Dementia-Specific Updated Training Programs

Family physicians or general practitioners are often the first line of contact for most people with mild cognitive impairment (MCI), an early sign of AD, or other dementias [66]. However, general practitioners are able to identify less than half of all people with MCI and are very poor at recording this in medical notes [67]. Thus, it remains unclear whether general practice physicians and nursing staff are prepared to diagnose and manage patients with dementia, rather than refer to specialists. This may reflect that they are unprepared, unconfident, or reluctant to see such patients in their clinic [68]. In this study, HCPs showed moderate proficiency on seven measures of knowledge regarding AD and dementia, with a mean score of 5.3 ± 1.8 out of 7 (~75%), with physicians displaying the highest proficiency, followed by students, educators, researchers, nurses, and other professions. Differences amongst stakeholders may be attributable to experience in practice and type of dementia-specific training received. Knowledge of the recent advances in basic pathophysiology were generally poor among all groups.

Next, given that there is a gap in knowledge and competence of certain HCPs towards dementia care, it is important to know of their attitude and willingness to improve. In a study of community health service centers in Beijing, China, Wang et al. [69] reported that general practitioners demonstrated limited levels of dementia knowledge and skills but expressed positive attitudes [69]. Primary care physicians from Quebec, Canada have displayed positive attitudes towards providing dementia care and expressed interest in more support and staff [68]. Dementia care is the responsibility of a multidisciplinary team of professionals. Bryans et al. [70] showed that whilst primary care nursing staff in central Scotland and London had a high level of knowledge on management strategies, they had lower proficiency on the epidemiology and diagnosis of dementia and hence lacked confidence in identifying dementia and managing coexisting behavioral and mental health challenges [70]. Vafeas et al. [33] quantitatively surveyed a group of 85 healthcare workers in Australia and found that although the majority have strongly positive views about people with dementia, a large number reported being afraid of such patients. Smyth et al. [29] in Queensland, Australia, showed that AD knowledge levels varied significantly between professional groups based on experience of caring for affected patients and having dementia-specific training [29].

These studies highlight the need for dementia-specific updated training programs for primary care practitioners to optimize care outcomes in aging populations. Liu at al. [71] report that dementia-trained physicians had significantly greater confidence and less negative views towards dementia care compared to non-trained physicians in Hong Kong, China. Hobday et al. [72] from Minneapolis, USA revealed that an online, 4-module dementia training program for nursing assistants and allied hospital workers significantly increased dementia care knowledge and was perceived to be useful, acceptable, feasible, and efficient. Galvin et al. [73] showed that an educational program in 540 nurses and other direct-care staff improved knowledge and confidence for recognizing, assessing, and managing dementia for at least four months post-training. Lintern [74] reiterates that nursing and care staff with more positive and “helpful” attitudes towards people with dementia are more likely to engage in social activities with patients and are more likely to use higher quality indicators during physical care tasks with improved staff attitudes and the quality of dementia care [74]. In the present study, only 40% of physicians and 30% of nurses reported having a dementia/neurodegenerative disease-specific training in the last 2 years, which provides an opportunity to introduce effective CME training programs.

### 4.3. Does Lifestyle Shape the Brain?

In our study, around 9 in 10 respondents believed that lifestyle shaped the brain and that there was an association between sleep, physical activity, and cognitive function. This proportion was slightly greater among physicians, possibly reflecting differences in practice experience, or participation in continuing education programs. This translates into providing better recommendations to patients in not only management but also prophylaxis, especially with respect to a disease without many disease-modifying treatments. This is important considering that AD patients and their caregivers consider physical activity to be meaningful and possible despite dementia [75]. Barnes et al. [76] estimate that almost half of all AD cases worldwide are attributable to potentially modifiable risk factors, including unhealthy lifestyle and physical inactivity, which has direct relevance to Qatar with its high comorbidity of chronic diseases. Awareness of such risk factors amongst healthcare professionals may allow for more prompt intervention with slower progression of dementia and its complications [77].

In the US, Europe, and the UK, physical inactivity is the highest population-attributable risk factor for AD, attributable for about 21% of the risk and equating to 16.8 million cases. Various prospective studies have shown that even mild to moderate physical activity may reduce the risk of dementia and AD [78,79]. Erickson et al. [80] observed that exercise reduces hippocampal cortical decay in the elderly; active individuals had overall better health, larger hippocampi, and better spatial memory. Recent data suggest that higher levels of physical activity in cognitively normal elderly are associated with lower plasma levels of AD-involved biomarkers such as plasma Aβ_1−40_, Aβ_1−42_, and APP_669−711_ in *APOE* ε4 noncarriers [81]. However, most large-scale trials and prospective studies examining the effects of exercise as a management option for AD are plagued by methodological inconsistencies and bias [82]. Sleep disturbance is associated with an increased risk of cognitive impairment and development of AD pathology. Whilst AD itself may lead to sleep disturbances, modifying the sleep-awake activity has been shown to induce changes in the soluble cerebrospinal fluid Aβ and tau concentrations, suggesting a bi-directional relationship [83]. Physical activity could also moderate the association between sleep and cognitive function and sleep and Aβ, sleep duration and episodic memory, sleep efficiency and episodic memory, sleep duration and Aβ, and sleep quality and Aβ [84].

### 4.4. Recent Advances in Diagnostics and Treatment of Dementia and Alzheimer’s Disease

Approximately 87.8% of all respondents were optimistic that dementia research will improve the outlook for dementia patients, their caregivers, and families. Research on dementia is influenced by the specific regional burden and shaped by the significant variation in population demographics and size, poverty, conflict, culture, land area, and genetics [85]. Innovation in diagnostics and treatment is required to reduce the burden of disease for future healthcare systems. In contrast to clinical diagnoses that are complex and vulnerable to potential errors, detection of dementia using objective biomarkers of bodily processes are currently receiving increased attention and funding [14,86]. Qatar Foundation, through its research funding wing Qatar National Research Fund (QNRF), has funded various biomarker-focused research projects on dementia. Corneal Confocal Microscopy (CCM) is a rapid ophthalmic imaging technique first developed to detect neurodegeneration in diabetic neuropathy [87]. It has recently shown great promise to detect early neurodegeneration in MCI and dementia, even more reliably than magnetic resonance imaging (MRI) [14,88,89] and has been proposed as a tool to longitudinally measure and objectively assess the effectiveness of novel drug treatments in clinical trials [90]. CCM has been shown to have superior diagnostic capability for MCI compared to brain volumetry [91], and can predict progression from MCI to dementia, comparable to hippocampal and whole brain volumetry [92]. In light of these recent promising findings from Qatar, we sought to investigate whether HCPs in the region were aware of this upcoming modality. Only one in five (18.8%) respondents reported at least moderate awareness of CCM and its potential in diagnostics compared to traditional methods. The greatest awareness was among educators and researchers (31.4%), with lowest awareness among nurses (10.4%).

A total of 68.1% of respondents in our study believed that artificial intelligence (AI) has potential to be utilized for the care of patients with neurodegenerative diseases and related disorders such as dementia. However, only 31.9% were at least moderately aware of the applications of AI in daily life and 32.3% were moderately aware about its application in early disease diagnosis. The greatest awareness of AI was among physicians and students, followed by educators and researchers (Table 5). AI in conjunction with CCM has shown considerable promise in diabetic neuropathy [93,94]. Recent advancements have shown the promise of computer-aided diagnostic tools for AD diagnosis; the analysis of large demographic datasets allows for stratification of risk factors and improvement of personalized therapies [95]. In a recent landmark paper published in *The Lancet Digital Health*, researchers have shown the potential of a deep learning model to effectively detect AD-dementia with accuracies ranging from 79.6–92.1% based on retinal photographs, a prospect similar to CCM [96].

A total of 86.9% of respondents in our study believed that HCPs from most if not all fields should be aware of latest advancements in dementia research, signaling openness towards learning. Despite a high number of ongoing trials for disease-modifying therapy, between 1988 and 2017, no less than 146 drugs failed in dementia clinical trials [14,97,98]. Therapeutic agents were said to not be able to alter AD-associated dementia disease course despite a temporary improvement in symptoms. However, the phase 3 Clarity AD trial earlier this year showed that lecanemab, a humanized IgG1 monoclonal antibody that binds with high affinity to Aβ soluble protofibrils, reduced amyloid in early AD and slowed down cognitive and functional decline, albeit with greater adverse events compared to placebo [99]. This trial represents the first among a long pipeline of disease- modifying therapies in patients with AD and dementia.

### 4.5. Implications of Present Findings for Dementia in Qatar and the MENA Region

In our secondary comparison of survey responses stratified according to primary location of practice/work/study, we observed that, compared to those based outside Qatar, respondents from within Qatar reported notable beliefs regarding dementia. A higher proportion of Qatar-based participants believed that loss of memory and loss of financial and daily independence were expected in the elderly and did not require medical attention. This is of concern in relation to recognition of early dementia. A lower proportion of Qatar-based respondents also denied that AD may be a result of black magic or psychological distress and that they would not resort to alternative medicine, highlighting lingering superstitious beliefs still present in the region. Qatar-based respondents also reported lower knowledge and awareness of the recent advances in the pathophysiology of dementia, highlighting a perhaps outdated knowledge base. This is of concern in the face of an explosive increase in the population of patients with MCI or dementia in the country.

To provide context to the current findings, we reviewed the contemporary literature on studies in dementia care in Qatar. A 2011 prospective study of 1660 adult Qatari attending primary health care clinics revealed that 1.1% had dementia, but strikingly 52.6% of those aged 50–65 had dementia [100]. Thus, primary care physicians should be expected to detect dementia in every other patient over the age of 50 who walks through the door, which requires appropriate knowledge and attitude [100]. Hamad et al. [101] have shown that the major primary causes of dementia in Qatar were AD (29%), vascular dementia (22%), mixed dementia (15%), and Parkinson’s disease (6%). This is important considering that subtype dictates treatment. Another important takeaway from this study was that the primary reason for late help-seeking behavior among families was due to the misconception that forgetfulness and other associated symptoms of cognitive impairment were considered to be part of the physiological aging process [101]. These findings were reiterated in a 2016 study of the first 100 patients referred to a memory clinic in Qatar, where researchers showed that 36%, 25%, and 23% of referrals were diagnoses of vascular, Alzheimer’s, and mixed dementias, respectively [102]. Furthermore, highly prevalent comorbidities in this cohort were hypertension, diabetes, dyslipidemia, history of stroke, vitamin D deficiency, and anemia. Additionally, compared to non-Qataris, Qataris presented with more severe behavioral and psychological symptoms of dementia [102]. A 2008 retrospective analysis of 50 dementia patients from a cohort of 350 home care patients in Qatar showed that AD and vascular dementia (secondary to stroke) were both equally the most likely pathologies leading to dementia and indeed atherosclerosis and hypertension were significant comorbidities [103]. A 2014 retrospective analysis of 889 elderly patients from major geriatrics facilities in the country showed that more than one in four had dementia, with 72% having some form of vitamin D deficiency, along with other comorbidities [104]. A 2009 study investigating the nutritional status of 130 long- term care Qatari patients found that 20.8% had dementia and more than one in five had lost more than 10% of their admission weight six months into long term care, with around 40% being under the fifth percentile of body mass index, highlighting the absence of appropriate nutritional assessment and nutrition care [105]. In a 2015 study, the rate of potentially inappropriate prescriptions was high among home care elderly patients and this risk was twice as likely in patients with dementia [106]. In a study of 24 COVID-19 positive cases in a long-term care facility in Qatar, 57% of elderly patients had dementia and three patients who died had dementia and diabetes [107]. This is despite the excellent national response of Qatar to geriatric mental health during COVID-19, compared to other Arab countries in the MENA region [108].

Kane et al. [109] have recently systematically reviewed major themes surrounding studies reporting on the “invisible” caregivers, namely family members, domestic workers, and private nurses, involved in dementia care. Older persons in Qatari society tend to be cared for by an intergenerational extended family with support from private nurses and domestic helpers. The influential role of religious beliefs [110], family connections, and social cohesion cannot be underestimated with mental health care leading the way for alternate healing practices. A widely documented practice is that of the services of a traditional or religious leader reciting religious texts on the patient’s body to dispel the “evil eye” to relieve symptoms of dementia [111]. Research on informal caregivers with a focus on the specific environment surrounding dementia care within the Arab-Islamic sociocultural context is rare. However, one such study reveals that the care-giving experience intersects with various influences through numerable themes, among which social stigma, personal knowledge of ADRD, and socio-religious attitudes towards caregiving of older persons recurrently influence dementia care [112].

A 2020 collaborative effort of various experts in Qatar summarizes the challenges faced by an aging population with respect to effective mental health as: (a) lack of integration among providers; (b) absence of coordinated data management mechanisms and need for more evidence-based research; (c) lack of specialized human resources in geriatric psychiatry and social care; (d) context-specific sociocultural factors that inhibit help-seeking [113].

## 5. Limitations of the Study

The results of the study cannot be interpreted without understanding the potential limitations with it. One limitation is the low response rate which might not render the results of the study generalizable to all the population of potential participants that it was intended to. However, as a pilot study, its results can help shed light on the topic and results can help in the design of future studies. A second limitation is the small sample size that does not help detect statistically significant difference among groups where observed differences are sizable. A third limitation is that answers are self-reported and might not reflect proper knowledge of some topics. In future studies we recommend using case studies in order to ensure that knowledge might be correctly reflected. Fourth, although the questionnaire utilized in the survey was constructed using components of previously published and tested surveys, as well as collaboration with experts in the field, its reliability as a whole has not been tested, nor was it circulated in the Arabic language, which may have introduced unintended bias or contributed to the low response rate. A fifth limitation is that we utilized a CME distribution list which contained HCPs based both within and outside Qatar, and although this provided opportunity for a secondary analysis based on location, we are unaware of what fraction of Qatar-based HCPs were invited for the survey. These limitations fall under the umbrella of lessons learnt from a pilot study, and future projects are likely to improve on these drawbacks.

## 6. Conclusions

To the best of our knowledge, this is the first study investigating the knowledge, attitude, and awareness of various healthcare stakeholders, ranging from physicians, nurses, medical students, educators, researchers, and allied health professionals in Qatar. The results of this pilot study will be used for a community needs-based assessment of local healthcare professionals to estimate the feasibility, demand, and need for a continuing education program on neurodegenerative diseases, with a focus on dementia and AD. We show that the overall knowledge of HCPs on dementia and AD is moderate, and whilst their attitude is largely positive, their awareness of basic disease pathophysiology and recent research advances is lacking. Notable differences in relation to specific professions both within and outside Qatar merit further study. Measures to improve the care of patients with dementia should include improving knowledge, attitudes, and awareness, while reducing stigma and caregiver burden, by providing specialized education and training. This pilot program will be followed up with a large-scale cross-sectional analysis of both healthcare practitioners and the general public to establish more accurate data in specific groups of practitioners as well as to determine the focus of public awareness campaigns to improve awareness of risk factors, reduce stigma, and improve help-seeking behavior to improve the management of patients with dementia.

## Figures and Tables

**Figure 1 ijerph-20-04535-f001:**
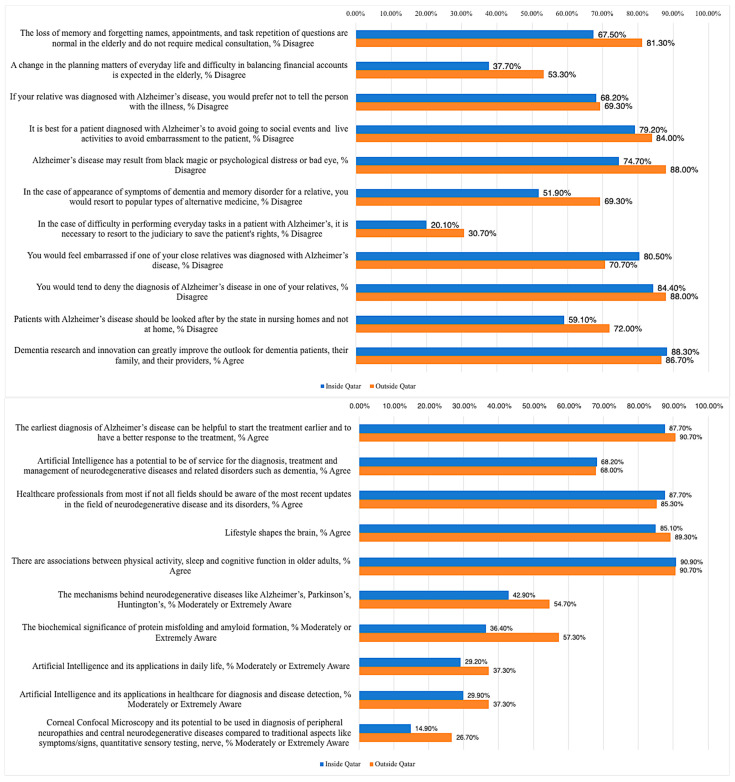
Comparison of responses from participants reporting their primary place of practice/work/study as Qatar (n = 154; blue) versus outside (n = 75; orange).

**Table 1 ijerph-20-04535-t001:** Qatar and its approach to dementia care.

Key Consideration	Description	References
Sociodemographic profile of Qatar.	Qatar is located halfway along the western coast of the Arabian Gulf and extends over an area of 11,627.8 square kilometers. The country is divided into eight municipalities with Doha being designated as the capital city, commercial center, and cultural hub where half the population lives. In 2020, the country’s population stood at 2.83 million, comprised of more than 80 nationalities. Qatar has a dry, subtropical desert climate, with little rainfall and extremely hot and humid summers. Islam is the official religion of the country; however, significant populations of followers of other religions live in the country. Arabic and English are widely used as the official and second language(s), respectively.	[16,17,18]
Qatar’s elderly population is growing rapidly.	In 2019, the population of elderly aged ≥ 65 years represented 1.5% (43,000) of Qatar’s population, compared to the world average of 9%. By 2050, this is expected to increase to 14.2% (546,000), signaling an explosive ~10-fold increase. In 2019, of a total of 2200 deaths; 912 (41.5%) were in persons aged ≥60 years and most deaths were a consequence of non-communicable diseases. Life expectancy at birth is 77.2 years, and at age 60, it is 19.2 years; however, the Healthy Life Expectancy at age 60 is 14.2 years, indicating the burden of morbidity in the elderly associated with chronic illness, cognitive decline, and functional dependency.	[7,18,19]
Healthcare services in Qatar.	Healthcare in Qatar is overseen by the Ministry of Public Health (MoPH) and is delivered primarily by public, semi-governmental services. In 2010, spending on healthcare accounted for 2.0% of the country’s GDP. The entire population has universal health coverage though public hospitals, although some employers also provide private hospital insurance for employees. Medicine is highly subsidized, with the exception of some highly specialized services. There are more than 7000 physicians, 2000 pharmacists, 9000 allied health professionals, and 22,000 nurses in Qatar, with 2.7 physicians and 8.1 nurses per 1000 people.	[16,18,20]
Dementia and health of the elderly population are recognized as national priorities in Qatar.	Unanimously adopted in 2017 by all member states, the World Health Organization (WHO) Global Action Plan on Dementia requires all nations to develop national dementia plans; however, only 40 countries have done so until now. In 2018, Qatar became the first Arab nation to launch a National Dementia Plan to build a pathway towards streamlined services for people with dementia, their families, and caregivers. It also published the five-year National Health Strategy 2018–2022, wherein one of seven focus areas included healthy aging of the elderly population. In January 2020, the MoPH published a comprehensive National Clinical Guideline on Dementia for the diagnosis and management of adults with dementia. Qatar Foundation, via the World Innovation Summit for Health (WISH) initiative, highlighted dementia on a global stage through publication of a “Call to Action” report at the conclusion of the WISH conference which had delegates from numerous WHO countries.	[4,14,15,18,21,22,23]

**Table 2 ijerph-20-04535-t002:** Demographic characteristics and comparison of survey respondents based on different occupational groups.

	Physician	Nurse	Student	Educators, Researchers	All Others	Total
**N (%)**	47 (20.5%)	48 (21.0%)	56 (24.5%)	35 (15.3%)	43 (18.8%)	229 (100.0%)
**Age range**						
1. 17–20	0 (0.0%)	0 (0.0%)	31 (55.4%)	0 (0.0%)	1 (2.3%)	32 (14.0%)
2. 21–30	4 (8.5%)	5 (10.4%)	25 (44.6%)	1 (2.9%)	7 (16.3%)	42 (18.3%)
3. 31–40	8 (17%)	22 (45.8%)	0 (0.0%)	10 (28.6%)	14 (32.6%)	54 (23.6%)
4. 41–50	15 (31.9%)	16 (33.3%)	0 (0.0%)	12 (34.3%)	12 (27.9%)	55 (24.0%)
5. 51–60	14 (29.8%)	5 (10.4%)	0 (0.0%)	3 (8.6%)	5 (11.6%)	27 (11.8%)
6. ≥61	6 (12.8%)	0 (0.0%)	0 (0.0%)	9 (25.7%)	4 (9.3%)	19 (8.3%)
**Primary location of practice/study**						
1. Inside Qatar	25 (53.2%)	35 (72.9%)	50 (89.3%)	15 (42.9%)	29 (67.4%)	154 (67.2%)
2. Outside Qatar	22 (46.8%)	13 (27.1%)	6 (10.7%)	20 (57.1%)	14 (32.6%)	75 (32.8%)
**Proportion of patients that are elderly (>60 years)**						
1. 0–10	14 (29.8%)	22 (45.8%)	32 (57.1%)	19 (54.3%)	15 (34.9%)	102 (44.5%)
2. 11–30	10 (21.3%)	4 (8.3%)	7 (12.5%)	2 (5.7%)	12 (27.9%)	35 (15.3%)
3. 31–50	7 (14.9%)	12 (25%)	14 (25%)	4 (11.4%)	8 (18.6%)	45 (19.7%)
4. 51–70	8 (17%)	7 (14.6%)	3 (5.4%)	5 (14.3%)	5 (11.6%)	28 (12.2%)
5. ≥70	8 (17%)	3 (6.3%)	0 (0%)	5 (14.3%)	3 (7%)	19 (8.3%)
**Exposure to patients with dementia**						
1. Never	4 (8.5%)	10 (20.8%)	27 (48.2%)	15 (42.9%)	14 (32.6%)	70 (30.6%)
2. Very Rarely	14 (29.8%)	21 (43.8%)	18 (32.1%)	9 (25.7%)	14 (32.6%)	76 (33.2%)
3. Rarely	8 (17%)	5 (10.4%)	5 (8.9%)	0 (0%)	4 (9.3%)	22 (9.6%)
4. Occasionally	5 (10.6%)	9 (18.8%)	4 (7.1%)	5 (14.3%)	6 (14%)	29 (12.7%)
5. Frequently	7 (14.9%)	2 (4.2%)	2 (3.6%)	3 (8.6%)	3 (7%)	17 (7.4%)
6. Very frequently	9 (19.1%)	1 (2.1%)	0 (0%)	3 (8.6%)	2 (4.7%)	15 (6.6%)
**Proportion with training in neurodegenerative disease or dementia in the last 2 years**	19 (40.4%)	15 (31.3%)	13 (23.2%)	12 (34.3%)	9 (20.9%)	68 (29.7%)

**Table 3 ijerph-20-04535-t003:** Knowledge of Alzheimer’s disease and dementia (n = 229). Figures represent correctness of answers, N (%).

	Physicians	Nurses	Students	Educators, Researchers	All Others	Total
	47	48	56	35	43	229
The loss of memory and forgetting names, appointments, and task repetition of questions are normal in the elderly and do not require medical consultation.	36 (76.6%)	35 (72.9%)	42 (75.0%)	26 (74.3%)	26 (60.5%)	165 (72.1%)
A change in the planning matters of everyday life and difficulty in balancing financial accounts is expected in the elderly.	27 (57.4%)	17 (35.4%)	21 (37.5%)	15 (42.9%)	18 (41.9%)	98 (42.8%)
Alzheimer’s disease may result from black magic or psychological distress or bad eye.	42 (89.4%)	38 (79.2%)	42 (75.0%)	28 (80.0%)	31 (72.1%)	181 (79.0%)
The earliest diagnosis of Alzheimer’s disease can be helpful to start the treatment earlier and to have a better response to the treatment.	41 (87.2%)	44 (91.7%)	49 (87.5%)	32 (91.4%)	37 (86.0%)	203 (88.6%)
Artificial intelligence has a potential to be of service for the diagnosis, treatment, and management of neurodegenerative diseases and related disorders such as dementia.	34 (72.3%)	28 (58.3%)	45 (80.4%)	19 (54.3%)	30 (69.8%)	156 (68.1%)
Lifestyle shapes the brain.	44 (93.6%)	37 (77.1%)	49 (87.5%)	32 (91.4%)	36 (83.7%)	198 (86.5%)
There are associations between physical activity, sleep, and cognitive function in older adults.	46 (97.9%)	41 (85.4%)	53 (94.6%)	31 (88.6%)	37 (86.0%)	208 (90.8%)
**Overall Knowledge (OUT OF 7)**						
**Mean ± SD**	5.7 ± 1.3	5.0 ± 1.5	5.4 ± 1.4	5.2 ± 1.4	5.0 ± 1.8	5.3 ± 1.5
**Median (Q1–Q3)**	6.0 (5.0–7.0)	5.0 (4.0–6.0)	6.0 (5.0–6.0)	6.0 (5.0–6.0)	5.0 (4.0–6.0)	5.0 (5.0–6.0)
**Min–Max**	3.0–7.0	0.0–7.0	0.0–7.0	1.0–7.0	0.0–7.0	0.0–7.0

**Table 4 ijerph-20-04535-t004:** Attitude on Alzheimer’s disease and dementia.

	Physician	Nurse	Student	Educators, Researchers	All Others	Total
**N (%) Disagree or Strongly Disagree**						
If your relative was diagnosed with Alzheimer’s disease, you would prefer not to tell the person with the illness.	33 (70.2%)	33 (68.8%)	39 (69.6%)	21 (60.0%)	31 (72.1%)	157 (68.6%)
It is best for a patient diagnosed with Alzheimer’s to avoid going to social events and live activities to avoid embarrassment to the patient.	39 (83.0%)	38 (79.2%)	47 (83.9%)	29 (82.9%)	32 (74.4%)	185 (80.8%)
In the case of appearance of symptoms of dementia and memory disorder for a relative, you would resort to popular types of alternative medicine.	34 (72.3%)	22 (45.8%)	34 (60.7%)	22 (62.9%)	20 (46.5%)	132 (57.6%)
In the case of difficulty in performing everyday tasks in a patient with Alzheimer’s, it is necessary to resort to the judiciary to save the patient’s rights.	12 (25.5%)	11 (22.9%)	14 (25.0%)	11 (31.4%)	6 (14.0%)	54 (23.6%)
You would feel embarrassed if one of your close relatives was diagnosed with Alzheimer’s disease.	30 (63.8%)	38 (79.2%)	50 (89.3%)	28 (80.0%)	31 (72.1%)	177 (77.3%)
You would tend to deny the diagnosis of Alzheimer’s disease in one of your relatives.	41 (87.2%)	40 (83.3%)	49 (87.5%)	31 (88.6%)	35 (81.4%)	196 (85.6%)
Patients with Alzheimer’s disease should be looked after by the state in nursing homes and not at home.	33 (70.2%)	24 (50.0%)	34 (60.7%)	2 (65.7%)	31 (72.1%)	145 (63.3%)
**N (%) Agree or Strongly Agree**						
Dementia research and innovation can greatly improve the outlook for dementia patients, their family, and their providers.	41 (87.2%)	43 (89.6%)	49 (87.5%)	31 (88.6%)	37 (86.0%)	201 (87.8%)
Healthcare professionals from most if not all fields should be aware of the most recent updates in the field of neurodegenerative disease and its disorders.	40 (85.1%)	42 (87.5%)	49 (87.5%)	31 (88.6%)	37 (86.0%)	199 (86.9%)

**Table 5 ijerph-20-04535-t005:** Awareness of pathophysiology and understanding of new advancements regarding neurodegenerative diseases.

Either Moderately OR Extremely Aware Regarding Following, N (%)	Physician	Nurse	Student	Educators, Researchers	All Others	Total
The mechanisms behind neurodegenerative diseases like Alzheimer’s, Parkinson’s, Huntington’s.	29 (61.7%)	18 (37.5%)	24 (42.9%)	18 (51.4%)	18 (41.9%)	107 (46.7%)
The biochemical significance of protein misfolding and amyloid formation.	26 (55.3%)	6 (12.5%)	32 (57.1%)	22 (62.9%)	13 (30.2%)	99 (43.2%)
Artificial intelligence and its applications in daily life.	19 (40.4%)	10 (20.8%)	24 (42.9%)	11 (31.4%)	9 (20.9%)	73 (31.9%)
Artificial intelligence and its applications in healthcare for diagnosis and disease detection.	17 (36.2%)	10 (20.8%)	25 (44.6%)	12 (34.3%)	10 (23.3%)	74 (32.3%)
Corneal Confocal Microscopy and its potential to be used in diagnosis of peripheral neuropathies and central neurodegenerative diseases compared to traditional aspects like symptoms/signs, quantitative sensory testing.	13 (27.7%)	5 (10.4%)	8 (14.3%)	11 (31.4%)	6 (14%)	43 (18.8%)

**Table 6 ijerph-20-04535-t006:** Summary of a literature review of contemporary studies from the Middle East and North Africa region investigating the knowledge, attitude, or awareness of the general public and healthcare practitioners regarding dementia and Alzheimer’s disease through quantitative survey methods.

Region, Country	Participant Demographics	Main Findings and Conclusions	Reference
**General Public**			
**Aseer, Saudi Arabia**	1373 participants, 769 (65%) females. Mean age 33 ± 11 years, 765 (55.6%) between 30–60 years. 808 (58.9%) married and 518 (36%) singles. 77% with no family history of AD. 1018 (74.1%) with education of bachelor’s or more. 1036 (75.4%) living with a small family.	<50% of questions were correctly answered. Mean ADKS knowledge score was 10.77 ± 5.1 out of 30. Younger females, and those with family history of the disease showed significantly (*p* < 0.05) better AD-related knowledge. More educational programs are required to improve public awareness and knowledge about AD.	[37]
**Jeddah, Saudi Arabia**	1698 individuals recruited in malls and public places, majority being between ages of 21–40 (55.9%). The majority were males (50.2%), Saudi nationals (75.1%), and residents of Jeddah city (94.3%). 74. 5% reported living with family, whereas 46.1% and 38% reported being single and married, respectively. 61.1% reported having a monthly income of ≤10,000 SAR or approx. 2700 USD; majority of participants were holders of bachelor’s degrees (33.6%) or high school students (30%). 28.4% reported having a personal experience with AD, involving a family member or friend.	89% had heard about AD. 44.9% believed that AD is a normal part of aging. Only 6.3% of knowledge about AD was from physicians. 46% incorrectly believed that AD causes hand tremor. 17.5% believed that AD can be treated by the recitation of Quran, showing high influence of cultural beliefs and religion. 45.8% believed that persons with AD are a burden on the family. 47.9% reported not knowing how to deal with persons with AD and 72.2% believed that persons with AD need monitoring throughout the day. Only 30% of the participants believed AD can be treated with medication. The authors conclude that awareness campaigns and public education are required to increase the knowledge of the general public regarding several aspects of AD which may lead to earlier detection and initiation of appropriate care.	[40]
**Makkah, Saudi Arabia**	575 ADKS survey respondents; majority of the respondents were female (65.2%), university graduates (67.7%), and between the ages of 18–25 years (84.3%). Only 23.1% had relatives diagnosed with AD. 68% and 22.8% of respondents reported getting their medical information from health care providers and media, respectively.	Mean ADKS score was 17.35 (out of 30), translating to 57.5%, which meant respondents were not regarded as appropriately knowledgeable. Female gender was associated with higher knowledge on Life Impact, Risk Factors, Assessment and Diagnosis, Caregiving, Treatment, and Management portions of the ADKS, as well as the overall score. Having a postgraduate qualification, higher age, relatives with AD diagnosis, and having newspapers and journal articles as a source of medical information were associated with higher knowledge on Caregiving, Assessment and Diagnosis, Risk Factors, and Life Impact sections of the ADKS, respectively. The authors recommend community campaigns to enhance AD awareness and continuing medical courses for health care providers.	[41]
**Beirut, Lebanon**	254 PHC attendees interviewed face-to-face. Majority of participants were female (73%), employed (58%), aged between 25–44 years (40%), and did not reach university (67%). 49% had prior personal exposure to persons with dementia.	Only 46% knew that dementia is not a normal process of aging. Only 20% knew that AD and dementia were not synonymous. Overall, 61% of participants had low knowledge score. In general, the elderly, those with low education level, and absence of previous personal exposure were predictors of low knowledge. The participants greatly agreed that those with dementia are helpless, dependent, deprived of their rights, and unable to make decisions on their own. 85% considered patients with dementia to be like children, needing similar care. Notably, more than 80% believed that “people who have just been diagnosed with dementia are unable to make decisions about their own care”, and not treated as human beings after diagnosis. Only half disagreed with the statement that life was not worth living for people with really bad dementia.	[42]
**Emeq Hefer, Israel**	Household representative sample of 1198 older adults with mean age 70.78 ± 9.64 years. Participants were divided into three ethnic groups: long term Israeli Jews (LTIJ), immigrants from the Former Soviet Union (FSU), and Palestinian Citizens of Israel (PCI).	Attitude towards AD was most accepting among PCI and most negative among FSU, whereas AD-preventative behaviors followed the opposite pattern. Demographics, human and economic resources, and AD familiarity reduced observed intergroup differences in AD-preventative behavior, but not on attitudes towards AD.	[43]
**Haifa, Israel**	185 adult child primary caregivers for parents with AD; 12.3 ± 19.2 h per week over 5.4 ± 4.1 years of caregiving. 74.6% female; mean age of 52.7 ± 8.8 years. 75.7% born in Israel, 73.5% married with 2.3 ± 1.2 children, 15 ± 2.6 years of education.	The two caregiver stigma variables of ‘shame’ and ‘decreased involvement with caregiving’ significantly influenced overall prediction of caregiver burden. Psychosocial interventions should target stigmatic beliefs in order to reduce caregiver burden, at least in those serving as family caregivers.	[44]
**Haifa, Israel and Hobart, Australia**	Israel: 447 adults; 50.1% female with mean age of 42.48 ± 13.10 years and 77.9% ethnical majority; 52.1% were familiar with someone with dementia. Age-matched Australian cohort: 290 adults; 50.2% females with mean age of 43.67 ± 14.21 years; 83.9% from ethnic majority; 36.6% were familiar with someone with dementia.	Israel vs. Australia; mean subjective knowledge score (1–5): 3.17 ± 0.92 vs. 2.57 ± 0.89 (*p* < 0.0001); mean agism (1–5): 2.05 ± 0.38 vs. 1.85 ± 0.42 (*p* < 0.001); mean dementia stigma (1–10): 2.44 ± 1.14 vs. 3.04 ± 1.26 (*p* < 0.001). Lower levels of subjective knowledge and higher levels of agism were associated with increased levels of stigmatic beliefs in both Israeli and Australian cohorts. Male participants reported higher levels of public stigma than women, although largely in Australia.	[45]
**Haifa, Israel**	514 layperson adults recruited via telephone call: 60.9% female with mean age of 53.4 ± 17.6 years and 14.3 ± 3.6 years of education. 72.6% were married, and 79.5% were Jewish; 48.1% were secular, while 41.9% were religious. 67.8% and 50.2% considered their health status and economic status to be excellent or good, respectively.	Gender, religiosity, and subjective knowledge of AD were statistically significant correlates of attitudes towards advance care planning. Knowledge of AD plays an important role in advance care planning attitudes.	[46]
**Haifa, Israel**	632 laypersons recruited via telephone call; mean age of 45 years and education of 14 years. The majority of participants were female (52.5%), Jewish (85%), and either traditional or secular (72%); 25.1% reported having a relative or an acquaintance with AD. Most were married (70%) and reported their income as equal to or above average (92%).	No gender differences in AD awareness were reported. Female participants consistently reported higher levels of perceived susceptibility, worry, fear, and knowledge about AD than males. Predictors of perceived susceptibility were familiarity with someone with AD, lower education, and higher age (only for females). Age and familiarity were predictors of AD knowledge among females, but only age for male participants.	[47]
**Riyadh, Saudi Arabia**	Female health care providers (n = 31), college teaching staff (n = 19), and students (n = 34).	Level of awareness increased with increasing age. Healthcare providers had outstandingly high level of awareness on the controllable risk of dementia compared to college students. The authors recommend efforts to increase awareness on the controllable risk factors for dementia, via campaigns or educational courses.	[48]
**Doha, Qatar**	2514 Qatari and non-Qatari Arabs visiting PHCs. 50.3% were Qataris and 49.7% were other non-Qatari Arabs. Most Qatari (54%) and non-Qatari Arabs (36.7%) belonged to the age group 31–45 years. More non-Qatari Arabs were males (58.6%) compared to Qataris (39.8%) More non-Qatari Arabs were university graduates (52.4%) and professionals (50.8%).	Compared to non-Qatari Arabs, a higher proportion of Qataris believed mental illness to be punishment from God (50.6% vs. 44.5%, *p* = 0.002) and that patients with such illness were mentally retarded (45.1% vs. 35.1%, *p* < 0.001). Qataris were more likely to believe that mental illness is caused by evil spirits (40.5% vs. 37.6%) and that psychiatric medication will cause addiction (61% vs. 57.3%). Qataris were less likely to recognize dementia than non-Qatari Arabs (3.1% vs. 4.4%). Overall, significant ethnic difference was observed in the knowledge, attitude, and practice of Qatari nationals versus Arab expats with regards to various mental illnesses. The authors acknowledge that steps to improve public perception of mental illness should be taken.	[49]
**Healthcare Professionals**			
**Tel Aviv, Israel**	197 family physicians, 49.2% male. Mean age 50 ± 9.2 years, range 28–69 years. Average seniority of 21.9 ± 10.4 years. 88% reported having at least a quarter of their patients being over 65 years or over with cognitive deterioration. 72.3% reported diagnosing a patient with MCI in the last month.	18% of all participants reported not having heard about MCI at all. A third of those familiar with MCI reported almost no subjective knowledge about it. 84.4% of participants believe family physicians can diagnose MCI. 71.7% of participants believed incorrectly that MCI is caused by normal aging; 84% did not believe that it was always pathological. Only 18.2%, 50.9%, and 59.1% correctly believed LBD, AD, and CBVD were causes of MCI, respectively. Despite low knowledge, willingness to learn about MCI was high (mean score of 4.1 ± 1.0 out of 5). Help-seeking and treatment preferences of participants agreed with the latest literature.	[50]
**Haifa, Israel**	327 nurses (205) and social workers (122) from medical centers and nursing homes. 87.5% and 94.3% of nurses and social workers were female, respectively. Mean age of nurses and social workers was 40.54 ± 10.15 and 41.33 ± 11.53 years, respectively. 49.5% and 79.7% of nurses and social workers reported being born in Israel, respectively. 75.1% and 64.8% of nurses and social workers reported being married, with an average of 2.23 ± 1.13 and 2.42 ± 1.01 children, respectively. Mean years of education were 15.51 ± 2.25 among nurses and 16.65 among social workers. 48.9% of nurses and 63.3% of social workers reported having an income lower than the national average.	Most AD symptoms were recognized by participants. Language difficulties and delusions were the only AD symptoms recognized by less than 60% of participants. Nurses attributed psychological reasons to AD more than the social workers. Surprisingly, social workers perceived AD to be more chronic with severe consequences compared to nurses. There are differences between the two groups regarding AD illness representations. Differences between groups in recognizing symptoms of AD were not present after adjusting for years of education. >50% of participants assumed that language difficulties and delusions are not related to AD. The authors recommend continuing to distribute materials to professionals regarding AD.	[51]
**Ramat Gan, Israel**	184 Filipino home care workers; mean age of 36.3 ± 8.3 years. Majority were not married (55.4%), female (83.7%), and reported having at least some college or trade school education (69.1%). Majority of the participants reported being in the country for between 1–5 years (70.7%) and working as a home care worker for the same period (70.7%). 16.3% reported never being informed about medical condition of their respective care recipient.	On 7 out of 14 items that assess ADRD knowledge, >30% of participants reported beliefs that were not parallel to majority culture and scientific literature. 51% believed ADRD was a normal implication of aging. 83% believed patients with ADRD needed constant supervision. 60% believed that it was favorable not to involve family members when the patient is agitated. Whether the home care giver was informed about the patient’s medical conditions was significantly (*p* < 0.005) associated with favorable/correct answers to the survey. In a sub-study using qualitative interviews, Filipino home care workers were knowledgeable about signs and symptoms of ADRD.	[52]
**Galilee region, Israel**	197 caretakers of dementia patients, belonging to 3 ethnic backgrounds: 36% Sabras, 41% Arabs, 23% Russians. Majority of respondents were nurses (55%) or nurse’s aides (31%). Majority had academic education (73.1%), female (76.02%), worked at nursing homes (55.84%), and had ≤10 years’ work experience (56.92%). Most reported being married (68.57%), and either secular (47.47%), religious (23.74%) or traditional (28.79%) with respect to religiousness.	Significant differences in nursing homes were found in the attitudes to autonomy and dignity of patients with dementia between Russian, Arab, and Sabra caretakers (index score for autonomy: 2.97, 4.07, and 4, respectively; index score for dignity: 3.17, 4.1, and 4.07, respectively). Most significant influencing variables on the indexes of autonomy and dignity were ethno-culture Arab/Russian and the patient’s family. Being a female also influenced the autonomy index. No demographic factor was associated with differential dementia attitude in hospitals.	[53]
**Turkey**	100 Turkish neurologists attending a neurology conference; 67% female with a mean age of 36.1 ± 8 years and mean 51.5 years of neurology expertise. 70% were married, with 49% having children and 78% having married parents. 93% had older adult relatives and 43% had a history of living with older adult relatives. 41% had a history of nursing home visits and 34% had participated in voluntary community activities. 27% had geriatrics education in medical school.	70% positive, 3% had neutral, and 20% had negative attitudes. Older participants and those with a history of geriatric education in medical school tended to have a higher rate of positive attitudes. Most negative items were regarding the natural course and behavior of the common diseases in practice. Interestingly, history of living with older relatives tended to have a negative effect on dementia attitude. The authors recommend generalization of geriatrics education to increase understanding and improve care for older patients with dementia.	[54]
**United Arab Emirates**	325 community pharmacists from three cities; 140 (Dubai), 170 (Sharjah), 15 (Ajman). Majority were males (54.8%), ≥30 years of age (54.8%), and of Indian nationality (52%). Most only had a bachelor’s degree (86.8%) and were ≤5 years into practice (58.2%). Most were employed full-time (94.5%) and worked at chain pharmacies (55.7%). Most spent ≤4 h per week on professional development (76%) and were dependent on internet websites for the same (88.9%).	Mean ADKS score was 57%; mean ADPM score was 68%. Overall, female pharmacists, pharmacists working for independently owned pharmacies, pharmacists ≥30 years of age, and those with >5 years of practice had significantly higher ADKS scores than their counterparts. History-taking, adherence assessment, and counseling were provided by only 2.2%, 9.3%, and 17.3% of respondents, respectively. Only 32.6% maintained a stock of AD management products. A minority provided high-quality counseling. The authors conclude that a multifaceted approach is required to develop patient-centered pharmacy-based services for Arabic-speaking communities.	[55]
**Riyadh, Saudi Arabia**	529 healthcare students enrolled at different colleges of King Saud University: pharmacy (223), medical (218), and dental (88). Majority of respondents were male (55%) and aged between 18–21 years (63.7%).	70.1% of students studied about AD in their college life. 70% and 73.5% could correctly recall that AD is related to mental disorders and is a neurodegenerative disease, respectively. The authors report significant difference in knowledge score of medical and dental students (*p* = 0.001) and that of pharmacy and dental students (*p* = 0.003). The authors believe that it is necessary to provide adequate education program and training in healthcare colleges to increase levels of knowledge.	[56]
**Jordan**	356 nursing students filling out DKAS-II; 31.74% and 38.2% of respondents were between the ages of 19–20 and 21–22 years, respectively. Majority of the respondents were female (72.75%), had a GPA between 3–3.49 out of 4.00 (46.35%), and were at the senior education level (36.52%). 88.2% had not taken a specialized course on dementia, whereas 53.65% had taken a course on mental health. 90.17% reported no family history of dementia, while 54.78% reported getting their information from the internet.	Mean DKAS-II score was 24.53 ± 7.81 out of 48 (52%). Lowest scores were on communication with and behaviors of people with dementia, and the risk factors and health promotion areas in dementia care. Students’ gender, current GPA, family history of dementia, and education level were significantly associated with mean total dementia knowledge and respective subscales (*p* < 0.05). The authors recommend raising the standards of dementia knowledge among future nursing professionals and implementing educational strategies in clinical settings that improve dementia care.	[57]

## Data Availability

Source data available on request from corresponding author.

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
