# Peer review of "Knowledge, Awareness, and Attitude of Healthcare Stakeholders on Alzheimer’s Disease and Dementia in Qatar"

_ijerph, 2023, doi:10.3390/ijerph20054535_

Round 1

Reviewer 1 Report

The authors have investigated a highly important topic regarding future developments in Qatar and other countries in the Middle East region. The study aim, design and results are clear. Overall, I believe that the the topic of this article is of interest and that it is also of interest to the readers of International Journal of Environmental Research and Public Health. However, I feel that the discussion section needs to be improved before publication.

Line 98: What is the background of these experts? How many experts? Could you be more specific?

Line 103: The questionnaire was not validated nor was its reliability tested. Could thus have influenced the results. Please explain.

Line 190: Even though, the questions are formulated neutrally, they are quite straight-forward. Participants might have felt obligated to provide socially desirable answers e.g., for example for question about black magic. Please elaborate.

Line 190: Several questions contain several options e.g., Artificial Intelligence has a potential to be of service for the diagnosis, treatment and management of neurodegenerative diseases and related disorders such as dementia. It is unclear to which of the points the participant agrees or disagrees.

Line 290: This paragraph is really long citing many previous studies without any connection to the results of the survey. Only from line 333 (reference to the Danish study), you use the previous study to place the results of the current study into context. In my opinion, this paragraph needs to be shortened.

Line 344 & 408: The same is true for the next two paragraphs. I miss a connection with the results of the study. Something like: Knowledge of the participating physicians shows gaps. Previous studies showed that this could be improved by dementia-specific updated training programs for primary care practitioners etc.

Line 425 & 457: This is what I also expect it the abovementioned paragraphs. Here you make the connection between the results and previous studies. However, I do feel that the description of previous studies is still too long. Line 494 does not add much to the previous section about AI. Therefore, I suggest to remove this section.

Line 503: I do not think that adding a review to the discussion section is appropriate. However, it adds context. Therefore, I would suggest to formalize the review (make it a systematic review with clear search terms) and add it as an additional research question to the study and add it the table to the result section as well as a description.

Line 511: Would not instead of wouldn’t

Line 586: Does not instead of doesn’t

Author Response

The authors have investigated a highly important topic regarding future developments in Qatar and other countries in the Middle East region. The study aim, design and results are clear. Overall, I believe that the topic of this article is of interest and that it is also of interest to the readers of International Journal of Environmental Research and Public Health. However, I feel that the discussion section needs to be improved before publication.

We thank the reviewer for their careful reading of our manuscript and acknowledgement of its importance in the region. We have carefully read and improved the discussion based on the specific suggestions provided below.

Line 98: What is the background of these experts? How many experts? Could you be more specific?

In addition to the authors, the survey was designed in collaboration with local and international researchers in the field of neurodegenerative diseases and physicians who routinely diagnose and manage such patients, in addition to institution-affiliated medical education experts who regularly design and deliver professional development programs and lectures for healthcare professionals in the region. We have included these specifics in section 2.1 Measures.

Line 103: The questionnaire was not validated nor was its reliability tested. Could thus have influenced the results. Please explain.

The reviewer is correct to point out that the questionnaire utilized in the current study was not tested priorly. However, as we have mentioned, the current study is a pilot that will help us to develop future questionnaires that better suit the population of interest; in a way, the current study is where its reliability is being tested. Additionally, as mentioned in section 2.1 Measures, the questionnaire (provided in supplementary materials) was developed in part from components of previously published and widely tested questionnaires in the field, including the Alzheimer’s Disease Knowledge Scale (ADKS), the Alzheimer’s Disease Aware-ness Scale (ADAS), and the Dementia Attitudes Scale (DAS), while region-specific components were later incorporated. Hence the results should be considered in the context of the questions and are unlikely to be biased hence. However, we have listed the above as a limitation to the current study in discussion.

Line 190: Even though, the questions are formulated neutrally, they are quite straight-forward. Participants might have felt obligated to provide socially desirable answers e.g., for example for question about black magic. Please elaborate.

Yes, indeed we have taken care to maintain a neutral tone in the survey questions. We agree with the reviewer that often-using straightforward language may deter respondents towards one answer or another, and this will be corrected for in future undertakings. However, our inspiration for the question was based on a similar awareness survey undertaken by Alhazzani et al. (https://doi.org/10.1186/s41983-020-00213-z) in Saudi Arabia, a culturally and demographically similar state to Qatar; here, despite the straightforward language, 35% of respondents believed that AD may result from black magic/psychological distress/bad eye, which was concerning, and we sought to determine whether this was true of healthcare stakeholders in Qatar. Indeed, only around 75% of medical students disagreed with the statement, which was an interesting finding with implications. Of particular interest in this case, Ghuloum et al. (https://doi.org/10.1192/S1749367600004975) reported that a widely documented practice in religious and conservative states like Qatar is to utilize the services of traditional/religious healers who recite religious texts onto the bodies of patients “to dispel the evil eye and relieve symptoms of dementia”. However, this is a practice perhaps in a minority of the general population. Regardless, we sought to understand if this notion was prevalent among HCPs. Similar rationales may be applied to other questions of the same nature. Nevertheless, we have listed the above as a potential limitation in the discussion.

Line 190: Several questions contain several options e.g., Artificial Intelligence has a potential to be of service for the diagnosis, treatment and management of neurodegenerative diseases and related disorders such as dementia. It is unclear to which of the points the participant agrees or disagrees.

We agree with the reviewer’s observation. Indeed, as a pilot study, we were limited with the number of questions permissible in the survey to stroke a balance between ensuring optimum number of responses and gathering meaningful data. However, to our knowledge, AI is not particularly being utilized for diagnosis/management of neurodegenerative disease and its complications in Qatar, hence our focus was rather to determine whether the respondents believed in the potential of AI to aid in any of these options, and not one particular.

Line 290: This paragraph is really long citing many previous studies without any connection to the results of the survey. Only from line 333 (reference to the Danish study), you use the previous study to place the results of the current study into context. In my opinion, this paragraph needs to be shortened.

We thank the reviewer for their suggestion. In this section we attempted to provide context regarding the perception of the general population towards AD/dementia around the globe, in order to later discuss similar results specific to the MENA region. After taking into the below suggestion of formalizing the review, we believe it will now better serve as an apt discussion. Nevertheless, we have shortened this section as suggested.

Line 344 & 408: The same is true for the next two paragraphs. I miss a connection with the results of the study. Something like: Knowledge of the participating physicians shows gaps. Previous studies showed that this could be improved by dementia-specific updated training programs for primary care practitioners etc.

We thank the reviewer for their suggestion. Similar to the above section, in this section we attempted to provide context regarding the awareness/attitude of towards AD/dementia around the globe, in order to later discuss similar results specific to HCPs in the MENA region. After accepting the suggestion of formalizing the current review, we believe it will now better serve as an apt discussion. We have also shortened this section and added the link suggested by the reviewer.

Line 425 & 457: This is what I also expect it the abovementioned paragraphs. Here you make the connection between the results and previous studies. However, I do feel that the description of previous studies is still too long. Line 494 does not add much to the previous section about AI. Therefore, I suggest to remove this section.

We thank the reviewer for their suggestion, which helped us understand the type of revision that is required. Regarding line 484, here we attempted to bring into context AD-modifying therapy research in order to discuss the responses measuring attitude towards AD research and recent advancements. We have therefore mentioned these prior to the section in question to provide a smoother transition.

Line 503: I do not think that adding a review to the discussion section is appropriate. However, it adds context. Therefore, I would suggest formalizing the review (make it a systematic review with clear search terms) and add it as an additional research question to the study and add it the table to the result section as well as a description.

We thank the reviewer for this important suggestion which aligns with feedback received from Reviewer 2. We have gone ahead and edited the methodology and results to formalize a part of this study as a formal review and appropriately replaced the tables.

Line 511: Would not instead of wouldn’t

Line 586: Does not instead of doesn’t

Thank you for your careful review, we have made the above changes and made similar revisions elsewhere.

Reviewer 2 Report

Dear authors,

I read with interest your manuscript. I think the topic is interesting. However, I have some major comments:

1. The survey was designed in English. It is not clear whether there was a translation in Arabic. I am aware English is also used in Qatar. But the lack of Arabic version could be a major limitation in the survey completion rate.

2. The survey has been sent to the mailing list of the Weill Cornell Medicine-Qatar continuing professional development division. Why this list was selected by the authors? What is the rate of HCP in Qatar who are registered in this list?

3. I am also confused by the presence of HCP who are not based in Qatar. Where is their working place? Are their answers relevant to develop an education program for HCP in Qatar?

Here are some minor concerns:

1. The introduction could be much more concise and more contextualised. The line 34- 48, 60 -79 are very general considerations, which could be clearly shortened. Instead of it, I appreciated the table 1 that provides more interesting information for this study. I would suggest the authors to transform it into the main text in the introduction.

2. The last table of the summary of similar studies in the region is also very interesting. As the authors mentioned it in the methods section, I think it should be part of the results, rather than discussion.

Author Response

Dear authors,

I read with interest your manuscript. I think the topic is interesting. However, I have some major comments:

  1. The survey was designed in English. It is not clear whether there was a translation in Arabic. I am aware English is also used in Qatar. But the lack of Arabic version could be a major limitation in the survey completion rate.

We thank the reviewer for their interest in the topic. Indeed, Arabic is the official language in Qatar, so we agree that a translated survey would be welcome. However, due to various constraints, this was not possible due to the pilot nature of the study. However, since the survey was circulated only amongst various healthcare stakeholders- physicians, nurses, medical students, researchers/educators, etc.- most of whom practice/work/teach/learn in English-medium settings, this is unlikely to be a major drawback for the study in question; nevertheless, we have listed this as a limitation in discussion as it could play a role in the low response rate. We have planned a larger survey to assess similar factors among the general population, for which we will undoubtedly prepare an Arabic version.

  1. The survey has been sent to the mailing list of the Weill Cornell Medicine-Qatar continuing professional development division. Why was this list selected by the authors? What is the rate of HCP in Qatar who are registered in this list?
  2. I am also confused by the presence of HCP who are not based in Qatar. Where is their working place? Are their answers relevant to develop an education program for HCP in Qatar?

Thank you for allowing us to explain these points in detail. The division of Continuing Professional Development (CPD) at WCM-Q is the department responsible for providing professional development and continuing medical education (CME) opportunities for physicians and other healthcare professionals in the region based on identified needs and the latest developments in an attempt to increase competence and improve patient outcomes. In this regard, the division maintains a list of HCPs who have either signed up to receive updates regarding various such opportunities, or who were enrolled from registration details of prior lectures; thus, although we expect the majority in the list to  be based in Qatar, we cannot exclude those based outside, nor can we estimate what fraction of Qatar-based HCPs are represented among the mailing list, which are limitations (now added in discussion). Available data regarding known professions and geographical location have been provided in Supplementary Materials. However, we may argue that the presence of a mixed cohort allowed us to consider our secondary analysis of comparing responses from HCPs based in Qatar versus outside, which is an added strength.

In a sense, the mailing list was a conveniently retrievable and existing resource available to us for this pilot study whose main objective was to conduct a needs assessment to identify the practice gap of HCPs in Qatar regarding AD/dementia. Indeed, over two-thirds of the respondents were based in Qatar, which allowed us to draw some conclusions based on response patterns, allowing us to conduct future surveys on more targeted populations with the experience of the current project. Although the responses of HCPs based outside Qatar may not be pertinent to developing education programs for Qatar-based HCPs, it provides us with a comparison benchmark to assess existing differences in knowledge, awareness, attitude in local HCPs to their foreign peers. However, we agree with the reviewer that a more specific, targeted, and optimized survey is necessary to assess these outcomes in Qatar-based HCPs, which we are working on currently and will distribute via central channels to ensure appropriate representation of the HCP-type population (physicians/nurses/students/others).

Here are some minor concerns:

  1. The introduction could be much more concise and more contextualized. The line 34- 48, 60 -79 are very general considerations, which could be clearly shortened. Instead of it, I appreciated the table 1 that provides more interesting information for this study. I would suggest the authors to transform it into the main text in the introduction.

Thank you for your suggestion. We agree that these parts of the introduction are rather broad generalizations that introduce the reader to the subject, rather than have region-specific information. We have shortened the section to remain in context with the study. Additionally, we agree with the reviewer that table 1 provides deeper context to the study, however, we believe that placing these details in a “one-stop-shop” table would enable easier readability.

  1. The last table of the summary of similar studies in the region is also very interesting. As the authors mentioned it in the methods section, I think it should be part of the results, rather than discussion.

Thank you for this important suggestion which aligns with suggestions of Reviewer 1. We have reorganized this section with appropriately modified methodology section in order to transform this table into the results of a formal review and replaced it appropriately.

Round 2

Reviewer 1 Report

Dear authors,

Thank you for revising the article. I think the revisions sufficiently improved the article.

Reviewer 2 Report

Thank you for this revised version. I feel that all my question have been fully addressed.